# Short-Term Treatment with Rho-Associated Kinase Inhibitor Preserves Keratinocyte Stem Cell Characteristics In Vitro

**DOI:** 10.3390/cells12030346

**Published:** 2023-01-17

**Authors:** Vignesh Jayarajan, George T. Hall, Theodoros Xenakis, Neil Bulstrode, Dale Moulding, Sergi Castellano, Wei-Li Di

**Affiliations:** 1Infection, Immunity and Inflammation Research & Teaching Department, UCL Great Ormond Street Institute of Child Health, 30 Guilford Street, London WC1N 1EH, UK; 2Genetics and Genomic Medicine Research & Teaching Department, UCL Great Ormond Street Institute of Child Health, 20 Guilford Street, London WC1N 1DZ, UK; 3Department of Plastic Surgery, Great Ormond Street Hospital for Children, Great Ormond Street, London WC1N 3JH, UK; 4Light Microscopy Core Facility, UCL Great Ormond Street Institute of Child Health, 30 Guilford Street, London WC1N 1EH, UK; 5UCL Genomics, Zayed Centre for Research into Rare Disease in Children, 20 Guilford Street, London WC1N 1DZ, UK

**Keywords:** keratinocyte stem cells, ROCK inhibitor, single-cell RNA sequencing, gene therapy, Y-27632

## Abstract

Primary keratinocytes including keratinocyte stem cells (KSCs) can be cultured as epidermal sheets *in vitro* and are attractive for cell and gene therapies for genetic skin disorders. However, the initial slow growth of freshly isolated keratinocytes hinders clinical applications. Rho-associated kinase inhibitor (ROCKi) has been used to overcome this obstacle, but its influence on the characteristics of KSC and its safety for clinical application remains unknown. In this study, primary keratinocytes were treated with ROCKi Y-27632 for six days (short-term). Significant increases in colony formation and cell proliferation during the six-day ROCKi treatment were observed and confirmed by related protein markers and single-cell transcriptomic analysis. In addition, short-term ROCKi-treated cells maintained their differentiation ability as examined by 3D-organotypic culture. However, these changes could be reversed and became indistinguishable between treated and untreated cells once ROCKi treatment was withdrawn. Further, the short-term ROCKi treatment did not reduce the number of KSCs. In addition, AKT and ERK pathways were rapidly activated upon ROCKi treatment. In conclusion, short-term ROCKi treatment can transiently and reversibly accelerate initial primary keratinocyte expansion while preserving the holoclone-forming cell population (KSCs), providing a safe avenue for clinical applications.

## 1. Introduction

Cultured epidermal sheets generated from primary keratinocytes have been used in autologous epidermal sheet graft therapy for severe burns and chronic ulcers for more than three decades [1]. This approach has recently been combined with gene therapies for rare genetic skin diseases, in which *ex vivo* gene-corrected epidermal sheets generated from patients’ keratinocytes, including keratinocyte stem cells (KSCs), are cultured and grafted onto patients with severe skin damage [2,3,4]. *Ex vivo* gene-corrected epidermal sheet therapy has been applied for junctional epidermolysis bullosa (JEB), recessive dystrophic epidermolysis bullosa (RDEB), and Netherton syndrome (NS) [2,3,4,5,6,7]. However, the success of this therapy relies on efficient correction of mutations in patients’ keratinocytes and retention of gene-corrected KSCs during the culture of gene-corrected epidermal sheet. The latter particularly relies on the efficient expansion of keratinocytes in *in vitro* culture [5]. As the epidermal sheet culture starts with limited primary keratinocytes isolated from small skin biopsies, it remains a challenge to effectively expand primary keratinocytes without adverse effects on KSCs including their self-renewal potential.

A human skin biopsy comprises 1–10% of KSCs, but only 0.1–1% of KSCs survive in *in vitro* cultivation [8]. This is because dissociation of KSCs from their surrounding cells and extracellular matrix can trigger apoptotic cell death, known as anoikis [9]. During epidermal sheet culture, it is important to maintain the KSC population as only KSCs have self-renewal capacity to give rise to sufficient numbers of transient-amplifying keratinocytes (TAs) and all other forms of differentiated keratinocytes (TDs) to form an epidermal sheet [10]. Under Green’s keratinocyte culture condition with feeder cells, heterogenous primary keratinocytes can be grown *in vitro* and form three types of clones, known as holoclones, meroclones and paraclones based on their morphology, size, and growth potential [11]. Using a single cell cloning approach, it has been confirmed that holoclones have the highest proliferative capacity, with less than 5% of aborted colonies upon sub-cultivation, whereas meroclones and paraclones have 5–95% and >95% of aborted colonies upon sub-cultivation, respectively [12]. Holoclone-forming cells, therefore, have hallmarks of stem cells, while meroclone- and paraclone-forming cells contain high proportions of TAs and terminally differentiated cells (TDs), respectively.

Advances in culture techniques have allowed holoclones to be sub-cultured *in vitro* to produce a large number of keratinocytes. Epithelial sheets can be formed once these cells attach to each other and stratify. Since epidermal sheet gene therapy requires genetically modified KSCs to produce higher numbers of gene-corrected TA and TD keratinocytes for sheet formation, an effective expansion of holoclones containing keratinocyte stem cells from primary keratinocyte culture is a crucial point to ensure the success of epidermal sheet culture. The presence of anoikis in freshly isolated keratinocytes can, however, reduce the yield of holoclone-forming cells and unduly prolong the culture time of epidermal sheets. This can lead to premature exhaustion of holoclone-forming cells and/or the loss of KSC stemness, resulting in a short-lived epidermal sheet graft gene therapy due to lack of holoclone-forming cells/KSC [5].

The discovery that rho-associated kinase inhibitor (ROCKi) helps evade anoikis-induced apoptosis in cultured human embryonic stem cells could be a potential solution [13,14]. ROCKi has been reported to prevent anoikis in embryonic, pluripotent, and prostate stem cells by activating protein kinase B (AKT) and inhibiting caspase 3 [13,15,16,17]. ROCKi could also induce proliferation and growth of hair follicle stem cells and astrocytes through the activation of extracellular signal regulated kinase (ERK) signalling [18,19]. It has therefore been widely used in somatic stem cell culture including limbal epithelial cells [20], pluripotent stem cells [21], endothelial progenitor cells [22], and KSCs [23]. In addition, a clinical trial is being conducted where ROCKi is administered along with human cultured corneal endothelial cells to treat corneal edema (ClinicalTrials.gov Identifier: NCT05309135).

Two ROCK inhibitors have been developed: the isoquinoline-derivative fasudil and the synthetic compound Y-27632. Fasudil has been approved to treat brain haemorrhages in Japan [24] and has also been used in clinical trials for strokes [25], brain blood flow disorder [26], angina, and other cardiovascular disorders and ocular diseases [27]. Rho-associated kinase is an effector in the small GTPase Rho pathway. It belongs to the protein kinase A, G, and C family and has pleiotropic functions including the regulation of cellular contraction, division, polarity, motility, gene expression, and morphology [28]. These functions vary with cell type and status, making it necessary to assess the molecular consequences of ROCKi treatment on each cell type to ensure its biological role; for example, whether the ‘forced’ proliferation affects the characteristics of KSCs and exhaust KSC population.

We investigated these questions by culturing human primary keratinocytes including KSCs obtained from human interfollicular skin with Y-27632 (hereafter referred as ‘ROCKi’). Cultured primary keratinocytes were treated with ROCKi for six days and then compared to non-treated cells to examine the changes in cell growth, protein markers and transcriptomics at single-cell resolution. We subsequently withdrew ROCKi in cultures for six days and compared these cultures against non-treated cells in the same way. In addition, we investigated the possible signalling pathways associated with cell proliferation and differentiation following ROCKi treatment in primary keratinocytes.

## 2. Methods

### 2.1. Skin Samples and Primary Keratinocyte Culture

Fresh skin samples were processed within two hours for isolation and culture of keratinocytes as described by Di, Mellerio [29]. Briefly, subcutaneous and adipose tissue were carefully removed from skin samples and the rest of the tissue was cut into small pieces (2 × 2 mm) and incubated with 0.02 U/mL neutral protease in PBS (Nordmark Pharma GmBH, Uetersen, Germany, #N0002936) for 3 h at 37 °C to detach epidermis from the dermis. The epidermis was then transferred to 0.25% trypsin in 0.01% EDTA solution (Thermo Fisher Scientific, Horsham, UK, #25200072) for 5 min to dissociate the epidermis into single cells. The dissociated cell solution was neutralized with Green’s medium [29]. Cells in the suspension were pelleted. After discarding the supernatant, pelleted cells were resuspended and plated out at a density of 4 × 10^4^/cm^2^ in culture flasks containing lethally irradiated 3T3-J2 cells (i3T3) and cultivated in a humidified atmosphere at 37 °C with 10% CO_2_. Subconfluent cells were passaged and re-seeded at a density of 1.5 × 10^4^/cm^2^ with i3T3 in Green’s medium containing DMEM and DMEM/F12 in the ratio of 1:1 supplemented with 10% FCS (Labtech, Heathfield, UK, # FCS-SA), 50 U/mL penicillin and streptomycin (Gibco, Paisley, UK, # 15070063), 10 ng/mL EGF (Peprotech, London, UK, # AF-100-15-1000), 0.4 μg/mL Hydrocortisone (Sigma, Gillingham, UK, # H-2270), 5 μg/mL Transferrin (Sigma, Gillingham, UK, # T2252), 5 μg/mL Insulin (Sigma, Gillingham, UK, # I-5500), 2 × 10^−11^ M Liothyronine sodium salt (Sigma, Gillingham, UK, # T6397), and 1 × 10^−10^ M Cholera toxin (Sigma, Gillingham, UK, # C-8052). Freshly isolated cells (P0) or cells with passage 1 (P1) were used in this study. All keratinocyte cultures were co-cultivated with i3T3 at 3 × 10^4^/cm^2^ unless specified.

### 2.2. ROCK Inhibitor Y-27632, Treatment Regimen and Cell Sampling

The ROCKi inhibitor Y-27632 ((R)-(+)-*trans*-4-(1-aminoethyl)-N-(4-pyridyl) cyclohexanecarboxamide-2) was purchased from AdooQ Bioscience (#A1101, Irvine, CA, USA). A total of 10 mM stock solution was prepared by dissolving ROCKi powder in 100% dimethyl sulfoxide (DMSO, Merck Life Science, Poole, UK, #D8418) and filtered through 0.22 µm syringe filters. The stock solution was then aliquoted and stored at −20 °C for the use in cell culture. A 1:1000 dilution of the stock solution was used to prepare a final concentration of 10 µM in culture medium.

Freshly isolated primary keratinocytes (P0) were seeded in 10 cm dishes and cultured in the Green’s medium containing 10 µM of ROCKi for six days. The culture medium was changed at day 3, replenished with fresh ROCKi. Cells cultured in Green’s medium without ROCKi but containing 0.1% DMSO were used for controls and run with treated cells in parallel (Figure 1). Briefly, cells were seeded in triplicates for each group for assays including cell growth rate and morphology, mitochondrial mass, single cell RNA sequencing (scRNAseq), and immunoblotting. Six days after seeding, cultured cells from both ROCKi-treated (ROCKi-6D^+^) and non-treated (Control-6D^−^) groups were harvested using 0.05% trypsin in DPBS. Before trypsinization, feeder layer was removed using 0.01% EDTA in DPBS for 30 s at room temperature. Trypsinized cells were neutralized with culture media and centrifuged at 500× *g* for 5 min at 4 °C. Cell pellets were resuspended in 0.5% fetal calf serum (FCS) in DPBS, and an aliquot was immediately proceeded for cell counting and cell size determination using the automated cell counter (CellDrop BF, DeNovix Inc, Wilmington, DE, USA). Half of the cells were harvested at day 6 (ROCKi-6D^−^) and divided into three portions for mitochondrial mass assay, single-cell RNA sequencing analysis and immunoblotting. The remaining half cells were seeded back in triplicates at a density of 1 × 10^6^ cells per 10 cm dish and grown for a further six days without addition of ROCKi, and then were harvested for analysis. Cells without ROCKi (Control-6D^−^6D^−^ and ROCKi-6D^+^6D^−^) were processed in the same way.

For short-time point experiments, P0 keratinocytes were seeded at a density of 1 × 10^6^ cells per 60 mm dish in the Green’s medium and cultured for two days. On day 2, culture media were changed with 3 mL of fresh Green’s medium. On day 4, 1 mL of 40 µM ROCKi in Green’s medium was added to the keratinocytes culture already containing 3 mL of culture media so that the final concentration of ROCKi was 10 µM in the culture. Control groups were treated with DMSO in parallel. These cells were incubated for either 15, 30, 60, or 120 min and were immediately rinsed with ice-cold PBS at the end of each time point and lysed directly on the culture plate using 1x Laemmli sample buffer supplemented with fresh 50mM DTT for the detection of AKT1/2 and ERK1/2 signalling.

### 2.3. Keratinocyte Growth Rate

Growth rates of keratinocyte cultures were calculated as previously described [30]. Growth rate was calculated using the equation, growth rate = ln (total number of cells harvested/total number of cells seeded)/duration of culture.

### 2.4. Colony-Formation Assay

The colony forming efficiency (CFE) of the keratinocytes was determined by a colony formation assay as previously described [31]. Keratinocytes were seeded in 10 cm dishes at a density of 800 cells per dish. On day 6 and day 12, dishes cultured with cells were washed twice with DPBS and stained with 0.25% rhodanile blue (Merck Life Science, Poole, UK, #121495) in 1% sulphuric acid/DPBS solution for 20 min at room temperature. Dishes were washed several times with DPBS and air dried. Colony staining image were scanned at the setting of 600 DPI using an HP Envy 4500 (HP, Bracknell, UK) scanner. A ruler was scanned at 600 DPI to measure the number of pixels per millimetre. Colony numbers and areas were counted and measured using Fiji software (https://imagej.net/software/fiji/, accessed on 22/04/2022). Briefly, images were converted to 8-bit grayscale. Uneven brightness of the staining was adjust using adaptive median filter (r = 2), the threshold was set to default, hollow colonies were filled by using binary fill holes, and joined colonies were separated using binary watershed. Colonies ≥ 1mm^2^ in area with the circularity of ≥0.5 were counted.

### 2.5. Immunoblotting

All antibodies used in this study are listed in Appendix A.

Keratinocytes were lysed directly on the culture plate using 1x Laemmli sample buffer (BioRad, Herts, UK, #1610747) supplemented with fresh 50mM DTT at room temperature. Cell lysates were passed through 20 gauge needle several times and then heated at 96 °C for 5 min. Total protein concentration in each sample was quantified using Pierce 660nM protein assay reagent (Thermo Fisher Scientific, Woolwich, UK, #22662) supplemented with ionic detergent compatibility reagent (IDCR, Thermo Fisher Scientific, Hampshire, UK, #22663). Equal amounts of total proteins were loaded in each lane, separated on Any kD Mini-PROTEAN TGX Stain-Free Protein Gels (BioRad, Herts, UK, #4568126) and transferred on to 0.22µm Nitrocellulose membrane using Trans-Blot Turbo semi dry transfer system (BioRad, Herts, UK). The transferred membranes were blocked for 1 h with 5% *w/v* non-fat milk in tris-buffered saline containing Tween 20 (TBST, 19mM tris base, 137mM NaCl, 3mM KCl, 0.2% Tween 20, pH 7.4) for 1 h and then incubated with primary antibodies diluted in blocking solution overnight at 4 °C. Membranes were washed three times (ten minutes each) in TBST before the addition of HRP-conjugated secondary antibodies for 2 h at room temperature. Membranes were washed three times (ten minutes each) in TBST, and target proteins were detected by using ECL Prime Western Blotting detection kit (SLS, Wilford, UK, # RPN2232). Signal was visualized and recorded using ChemiDoc MP imaging system (BioRad, Herts, UK). The membranes were re-used by stripping the previous antibodies using Restore Plus western blot stripping buffer (Thermo Fisher Scientific, UK, #46430) and re-probed for the house-keeping protein glyceraldehyde 3-phosphate dehydrogenase (GAPDH) as a loading control. Protein bands were quantified against GAPDH by optical densitometry using Image Lab 6.1 software (BioRad, Wilford, UK).

### 2.6. MitoTracker Green Staining

Primary keratinocytes were loaded with 200nM MitoTracker Green FM (Thermo Fisher Scientific, Horsham, UK, #M7514) and incubated at 37 °C for 30 min. Cells were then washed with DPBS, and feeder cells were removed using 0.01% EDTA in DPBS for 30 s at room temperature. Cells were then harvested by 0.05% of trypsin in DPBS, pelleted and then re-suspended in ice-cold 0.2% FCS in DPBS for flow cytometry analysis using CytoFLEX S (Beckman Coulter, Bucks, UK). Cells were excited at 488 nm and the emission fluorescence was collected at 535 nm. A total of 40,000 live cells were analysed per sample.

### 2.7. 3D Organotypic Culture

A 3D organotypic culture was performed as described by Di, Larcher [32]. A de-epidermalised dermis (DED, Euro Tissue Bank, BEVERWIJK, Netherlands) was used as a scaffold for generating 3D organotypic culture. A metal ring with 1 cm diameter was placed on the dermal side of the DED in a 35 mm dish and 2 × 10^5^ primary human dermal fibroblasts were seeded on the reticular side and cultured in the Green’s medium. After 24 h, the metal ring was removed, DED was flipped over (in the papillary side), and the metal ring was placed again on the papillary side. Primary keratinocytes sized 5 × 10^5^ were then seeded on the papillary side of the DED in Green’s medium. After 48 h, the metal ring was removed, and the DED culture was air-lifted by transferring it on to a mesh so that reticular side of the DED was in contact with the culture medium and the papillary side was in contact with air. Air-lifted DED with cells were cultured for 14 days, allowing the keratinocytes to proliferate and differentiate. Primary keratinocytes from three different donors were used to generate three biological replicates. Primary human fibroblast (passage 3 to 4) from same donor was used for all 3D cultures.

### 2.8. Cryosectioning and Immunostaining

Appendix A.

### 2.9. Single-Cell Library Preparation and RNA Sequencing

Cultured keratinocytes were harvested and resuspended in ice-cold buffer containing 0.5% FCS in DPBS. A small aliquot of the cell suspension was used to determine the percentage of live/dead cells by stained with Acridine Orange/Propidium Iodide (AO/PI) Cell Viability Kit (Vita Scientific, Beltsville, MD, USA, #LGBD10012) and counted by using LUNA-FL automated cell counter (Logos Biosystems, Annandale, VA, USA). Approximately 3000 live cells from each group were loaded into one channel of the Chromium Chip G using the Single Cell reagent kit v3.1 (10X Genomics, Pleasanton, CA, USA) for capturing single-cell Gel Bead Emulsion using the Chromium controller. Following single-cell capture, cells were lysed, and cDNA of individual cells were synthesized within the Gel Bead Emulsion and were used for library preparation. cDNA was amplified and size-selected using SPRISelect (Beckman Coulter, Krefeld, Germany, #B23317) following the manufacturer’s protocol. In all, 25 ng of the size-selected cDNA of each sample was used to construct Illumina sequencing libraries. Libraries were pooled at equal concentration and sequenced on the NovaSeq SP100 or NextSeq2000 Illumina sequencing platform at 50,000 reads per cell.

### 2.10. Generation of Gene Expression Matrix of Counts and Quality Control for scRNAseq

The gene expression count matrix was generated from scRNAseq raw sequences using cellranger [33] with the standard cellranger pipeline consisting of the mkfastq, count, and aggr steps. Gene expression was analysed using Seurat [34]. Genes present in fewer than three cells and cells expressing fewer than 200 genes were excluded from the analysis. Low-quality cells and barcodes corresponding to multiple cells or empty droplets were removed by excluding cells associated with either an anomalous number of genes, number of molecules, or percentage of mitochondrial DNA. Preparation of the Gel Bead Emulsion for each sample in separate wells caused these measures to vary between each other. Hence, we used different thresholds for each sample as a universal threshold would include low-quality cells in some samples and exclude high-quality cells in others (Appendix A). Non-keratinocytes were removed by excluding cells expressing (>1 count) of markers of melanocytes (*MLANA, PMEL, MITF*) and mesenchymal-like cells (*ACTG2, DLK1, MTRNR2L6, MTRNR2L10, MTRNR2L7, MTRNR2L1, PRKAR2B, NR2F1*) [35].

### 2.11. Normalisation and Cell Cycle Removal

Gene expression was normalised using sctransform [36]. The effect of cell cycle was mitigated by assigning scores to each cell representing the probability of its being in different stages of the cell cycle (using Seurat’s CellCycleScoring function) and then regressing out the difference between these scores using sctransform (satijalab.org/seurat/archive/v3.1/cell_cycle_vignette.html-“Alternative Workflow,” accessed on 18 May 2021).

### 2.12. Dimensionality Reduction and Donor Effect Removal

Dimensionality reduction was conducted by principal components analysis (PCA) using the 3000 most highly variable genes as features. The 50 most significant components were used since there was no clear point at which the generated components became less significant (Appendix A). Inter-donor variation was removed with Harmony [37].

### 2.13. Differential Expression and Differential Abundance Analyses

Seurat was used for differential expression analysis of treated and control cells with its standard Wilcoxon rank sum test. We used DAseq [38] to measure the proportion of two cell populations that cluster separately (detailed in Appendix A

### 2.14. Cell Type Proportion Comparison

SCINA [39] was used to compare the cell type proportions in the control and ROCKi-treated cells. Cell type markers published by Enzo, Secone Seconetti [40] were used to classify cells as either holoclone-forming, clonogenic, or terminally differentiated. Bootstrap resampling was used to estimate confidence intervals around the total absolute differences between cell type proportions.

### 2.15. Trajectory Analysis

Slingshot [41] was used to infer the differentiation trajectory for each sample class with the Harmony reduction. The groups of non-treated and ROCKi-treated cells were combined for the inference procedure to ensure that the pseudotimes were directly comparable. Three clusters were used for each sample (approximately corresponding to holoclone-forming, clonogenic, and terminally differentiated cells) as the input to Slingshot. The clusters likely to contain mostly holoclone-forming cells were used as the starting point for each inferred trajectory. Trajectories were then compared by computing the 0.1th, 0.2th, …, 99.9th, 100th percentiles of the pseudotimes corresponding to each sample and calculating the mean absolute difference at each percentile. Bootstrap resampling was used to estimate confidence intervals around the mean absolute differences between the pseudotimes both during ROCKi treatment and following its withdrawal.

### 2.16. Statistics

Experiments were performed in three individual donors (*n* = 3). Data are presented as mean ± S.D. Statistical analysis was performed using a Student’s *t*-test (two-sample, two-tailed, unequal variance, *p*  <  0.05 was considered significant).

For the single-cell RNAseq analyses, the state-of-the-art statistical analysis tool DAseq was used to identify the overall similarities between samples, and the Wilcoxon Rank Sum test was used to identify differentially expressed genes. Bootstrap resampling was used to create confidence intervals around the differences in cell type proportions differentiation behaviour between treated and control samples, and confidence intervals was used to show that the difference is likely to be at least 9 times larger during treatment than afterwards, which is a statically stronger claim than simply testing for a significant *p* value.

## 3. Results

### 3.1. Accelerated Colony Formation in the Culture with Six-Day ROCKi Treatment Stopped after ROCKi Withdrawal

Freshly isolated keratinocytes without passage (passage zero, P0) containing heterogenous cell populations including KSCs were cultured in Green’s keratinocyte culture medium with feeder cells and 10 µM of ROCKi for six days. The culture media were changed at day 3 and replenished with fresh ROCKi. At day 6, half of the cells were harvested for evaluation (ROCKi-6D^+^) while the remaining cells were continuously cultivated without ROCKi for further six days (ROCKi-6D^+^6D^−^). Cells without ROCKi treatment but cultured with 0.1% DMSO were used as negative controls (Control-6D^−^ and Control-6D^−^6D^−^, respectively) (Figure 1).

Colony forming efficiency (CFE) was assessed based on the size and circularity of colony. A 15-fold increase in colony formation was detected in ROCKi-6D^+^ cells compared to Control-6D^−^ cells (Figure 2a,b, *n* = 3, *p* < 0.05). The growth rate of keratinocytes was calculated and there was a fourfold increase in ROCKi-6D^+^ groups compared to Control-6D^−^ groups (Figure 2d,e, *p* < 0.05). The size of ROCKi-6D^+^ cells were smaller than those in Control-6D^−^ (Figure 2c, *p* < 0.05). Control-6D^−^ keratinocytes formed tightly packed colonies during the first two days of culture, whereas ROCKi-treated keratinocytes (ROCKi-6D^+^) dispersed with loose connections in the first 2–3 days culture but formed tightly packed colonies at day 4 (Appendix A). These morphological changes suggested that cells treated with ROCKi were in a proliferating state.

Since the mitochondrial content in stem cells is lower when they are dormant, but increases upon activation and proliferation, mitochondrial mass/content were examined using Mitotracker Green assay. Two distinct cell populations with low and high mitochondrial mass/content were observed in both ROCKi-6D^+^ and Control-6D^−^ cells. However, ROCKi-6D^+^ cells had a larger population with high mitochondrial mass than Control-6D^−^ cells (31.84 ± 1.99% v 3.56 ± 3.53%, *p* < 0.05) (Figure 2f,g). Live cell staining with Mitotracker green showed intensely stained cells located at the periphery of the colonies in which the proliferation marker keratin 14 (K14) was also highly expressed (Appendix A), suggesting elevated proliferation of keratinocytes in the culture treated with ROCKi.

Protein markers related to proliferation including K14, Integrin-α6, Integrin-β1, Aurora Kinase B (AUKB), HMGB2, FOXM1 and Δ-P63α, and differentiation marker K10 were examined. Proliferation markers, other than AUKB, FOXM1, and K14, were significantly increased in ROCKi-6D^+^ cells compared to Control-6D^−^ cells. The protein level of AUKB was significantly reduced, while FOXM1 and K14 remained unchanged. In contrast, the differentiation marker K10 was significantly lower in ROCKi-6D^+^ cells compared to Control-6D^−^ cells (Figure 2h, *p* < 0.05).

Importantly, in cells treated with ROCKi for six days and then passaged and cultured without ROCKi treatment for six additional days (ROCKi-6D^+^6D^−^), the colony formation efficiency, cell growth rate, cell size and mitochondrial mass returned to the levels comparable to those of non-treated cells (Control-6D^−^6D^−^) (Figure 3a–g). The markers related to cell proliferation and differentiation were also similar between the ROCKi-6D^+^6D^−^ and Control-6D^−^6D^−^ cultures (Figure 3h), indicating a transient effect of ROCKi on cells following short-term treatment.

### 3.2. Keratinocytes Treated with Short-Term ROCKi Maintained Differentiation Ability

The differentiation potential in cells with and without ROCKi treatment were evaluated in the organotypic culture *in vitro*. Fully differentiated multi-epidermal layers including a basal layer containing cube-shaped basal cells and suprabasal layers with stratum spinosum, granulosum, and anucleated cornium were detected in the cultures generated using Control-6D^−^, ROCKi-6D^+^, Control-6D^−^6D^−^, and ROCKi-6D^+^6D^−^ cells (Figure 4). This suggested that short-term ROCKi treatment did not alter the differentiation ability of keratinocytes. The expression and distribution of protein markers related to basal cells (Δ-P63α) and differentiated cells (K10, involucrin, and filaggrin) were examined. There were no differences in the expressions and distributions of markers of differentiated cells (K10, involucrin and filaggrin) (Figure 4k–y), but the expression of the basal cell marker Δ-P63α increased in ROCKi-treated cells (ROCKi-6D^+^) compared to non-treated cells (Control-6D^−^) (Figure 4g,h). However, the expression of Δ-P63α became indistinguishable between cultures withdrawal of ROCKi treatment (ROCKi-6D^+^6D^−^) and non-treated cells (Control-6D^−^6D^−^), indicating the enhanced proliferation cells did not change cell differentiation potential (Figure 4i,j). As Δ-P63α is a proliferation marker for basal keratinocytes (which include KSCs and transient amplifying cells [42]), the increase in Δ-P63α expression in the 3D organotypic culture generated using cells treated with ROCKi suggests that KSCs and transient amplifying cells were actively transitioning from quiescent to proliferative status in ROCKi-treated cells. ROCKi thus accelerates keratinocyte growth.

### 3.3. Transient and Reversible Changes in Transcriptome in Keratinocytes following Withdrawal of ROCKi Treatment

The transcriptomes in cells with 6-day ROCKi treatment (ROCKi-6D^+^), ROCKi withdrawal (ROCKi-6D^+^6D^-^), and non-treatment (Control-6D^−^, Control-6D^−^6D^−^) were examined by single-cell transcriptomic analysis (scRNAseq). Melanocytes were identified by the markers *MLANA, PMEL,* and *MITF,* and fibroblasts identified by the markers *ACTG2, DLK1, MTRNR2L6, MTRNR2L10, MTRNR2L7, MTRNR2L1, PRKAR2B, NR2F1* and further confirmed the expression of vimentin (*VIM*) (Appendix A). Both melanocytes and fibroblasts were excluded from transcriptomic analysis.

Individual keratinocytes that passed QC criteria were initially plotted in the uniform manifold approximation and projection (UMAP) based on the similarity of their transcriptomes. ROCKi-treated cells (ROCKi-6D^+^) and non-treated cells (Control-6D^−^) formed distinct clusters (Figure 5a), indicating large differences in transcriptomes between the two groups. In contrast, the cells following ROCKi withdrawal (ROCKi-6D^+^6D^−^) and non-treated cells (Control-6D^−^6D^−^) clustered together (Figure 5b), revealing the similarity of their transcriptomes. Quantification of the overlap of cell populations by DAseq [38] corroborated this observation, showing that only 10% of cells of ROCKi-6D^+^ clustered similarly to the cells of Control-6D^−^, whereas 63% of ROCKi-6D^+^6D^−^ overlapped with Control-6D^−^6D^−^ (Appendix A).

Fourteen differentially expressed genes were identified with most two-fold changes (log_2_ >1 or <−1) between ROCKi-6D^+^ and Control-6D^−^ (Figure 5c). Eleven genes (*VIM, KRT7, KRT19, IFI27, DCBLD2, F3, THBS1, IGFBP7, ARHGAP29, EMP1, FST*) were downregulated and three genes (*S100A8, FABP5, KRTDAP*) were upregulated. *VIM* is a fibroblast marker. As fibroblasts were excluded from the transcriptomic analysis, it was surprising to observe *VIM* expression in keratinocyte clusters. To confirm that the analysed data set did not include fibroblasts, the co-expressions of *VIM* and *K14,* which is a unique marker for keratinocytes, were analysed in the clusters classified as keratinocytes (Appendix A). The results confirmed that a proportion of keratinocytes co-express *K14* and *VIM*. This result was also consistent with other studies in which keratinocytes expressed *VIM* during wound healing and re-epithelization [43,44]. *KRT7* and *KRT19* are normally absent in skin, but in cultured keratinocytes, these keratins can present dependent on the culture conditions used [45]. *ARHGAP29* (which codes for Rho GTPase activating protein 29) regulates Rho GTPase signalling by interacting with Rap [46]. F3, also known as coagulation factor III, has been reported to highly express in the human epidermal skin during wound healing [47]. The genes *S100A8, FABP5,* and *KRTDAP* are associated with terminal differentiation of keratinocytes [48,49]. It is not clear why these gene expressions were changed in cells treated with ROCKi, but it was noticed that once short-term ROCKi treatment was withdrawn, there were no genes with at least a two-fold change between ROCKi withdrawn cells and non-treated cells (Figure 5d). This again indicated that the influence of short-term ROCKi treatment in cells was transient and reversible.

### 3.4. The Proportion of Holoclone-Forming Cells Was Reduced after 6-day ROCKi Treatment but Recovered following Its Withdrawal

As holoclone-forming cells have hallmarks of stem cells [12], the proportion of KSCs was evaluated based on the proportion of holoclone-forming cells. Single-cell transcriptomics was used to quantify the proportion of holoclone-forming cells, which were identified by the co-expression markers of *AURKB, CCNA2, CKAP2L, FOXM1,* and *HMGB2* described by Enzo, Secone Seconetti [40]. Whilst ROCKi-6D^+^ cells expressed higher levels of holoclone-forming cell markers, they had a reduced proportion of holoclone-forming cells compared to non-treated cells (Figure 6a,c, *p* < 0.05). On the other hand, in cells after the withdrawal of ROCKi treatment, the proportions of holoclone-forming cells were similar between ROCKi-6D^+^6D^−^ and Control-6D^−^6D^−^ (Figure 6b,c, *p* < 0.05).

The differentiation behaviour (from holoclone-forming cells to transient amplifying cells and finally to differentiated cells) was further analysed in treated and non-treated cells using single-cell trajectory inference (Figure 7a–d). The estimated pseudotime (differentiation progress) of each cell revealed that the progression from holoclone-forming cells to meroclone- and paraclone-forming cell types was significantly faster in ROCKi-6D^+^ cells than in Control-6D^−^ (Figure 7e, *p* < 0.05). However, this acceleration disappeared once ROCKi treatment was withdrawn (ROCKi-6D^+^6D^−^), with the differentiation rate in this culture similar to that of the culture without ROCKi treatment (Control-6D^−^6D^−^) (Figure 7f, *p* < 0.05). These results indicated that short-term ROCKi treatment accelerates the differentiation of holoclone-forming cells into meroclone-forming cells, but this acceleration was reversible once treatment was stopped. In addition, the acceleration from holoclone-forming cells to meroclone-forming cells during this short-term treatment was transient and may not exhaust holoclone-forming cells as the corresponding cell numbers were not changed after treatment was withdrawn (Figure 6b).

### 3.5. ROCKi Treatment Rapidly Activated AKT1/2 and ERK1/2 Signalling

It has been reported that Y-27632 treatment could suppress dissociation-induced apoptosis in murine prostate stem cells and iPSCs by inhibiting the activation of PTEN and also by preventing the deactivation of AKT, which, in turn, promotes anti-apoptotic and pro-cell cycle pathways through the putative PI3K-AKT-mTOR signalling [16]. Another study showed that U0126, an inhibitor of extracellular signal-regulated kinase 1/2 (ERK), could block Y-27632-induced cell proliferation in human hair follicular culture [19]. However, it is not clear whether ROCKi has these signalling influences in treated keratinocytes. Therefore, the activation of AKT1/2 and ERK1/2 was investigated. As ROCKi prevents dephosphorylation of myosin light chain (MLC2) by inhibiting the activity of MLC phosphatase [50], MLC2 phosphorylation in cells treated with ROCKi was assessed. Phosphorylated MLC2 was decreased within 15 min and at its lowest 30 min post-treatment (Figure 8a,b), indicating the impact of ROCKi in treated keratinocytes. Further, both AKT1 and AKT2 were activated with a significant increase in phosphorylated AKT1 (Ser473) and AKT2 (Ser474) within 15 min of treatment (Figure 8a). Finally, the phosphorylated states of c-RAF (Ser338), MEK1/2 (Ser217/221), and ERK1/2 (Thr202/204), which are involved in the ERK1/2 signalling pathway, were assessed. Cells treated with ROCKi significantly increased the phosphorylation of c-RAF and MEK1/2 in 15 min. However, the level of phosphorylation of ERK1/2 did not change (Figure 8a,b), suggesting a spatiotemporally regulated ERK signalling. Together, these results indicated that ROCKi stimulates primary human keratinocyte proliferation by ‘dual’ activation of AKT1/2 and ERK1/2 signalling pathways.

## 4. Discussion

Epidermal sheet gene therapies require *in vitro*-cultured primary keratinocytes containing genetically modified KSCs to produce enough proliferated and differentiated keratinocytes for epidermal sheet formation. *Ex vivo* epidermal sheet gene therapy has been applied for the treatment of rare genetic skin diseases. The long-term survival of genetically modified epidermal sheet therapy relies on the presence of genetically modified KSCs in *in vitro* cultured epidermal sheets. However, the stemness of KSCs can be affected by *in vitro* culturing environment. Freshly isolated KSCs could undergo dissociation-induced cell death (anoikis), causing low yield of KSCs or holoclone-forming cells. This could prolong *in vitro* cell cultivation to obtain sufficient keratinocytes for epidermal sheet formation. The prolonged epidermal sheet culture could eventually exhaust the stemness of KSCs and hinder the clinical application of epidermal sheet-based therapies. We thus investigated whether the use of ROCKi in initial primary culture could prevent these problems whilst preserving the holoclone-forming cell population and maintaining KSCs’ self-renewal properties and the ability of proliferation and terminal differentiation.

Previous studies of ROCKi treatment in keratinocytes isolated from human neonatal foreskin observed increased keratinocyte proliferation at the peak of day 6 with a 50-fold increase in colony forming efficiency [23]. A study has shown that long-term exposure (passage 122) of cultured keratinocytes to ROCKi increased proliferation without inducing any neoplastic transformation [51]. However, there were reports that more than a hundred-day treatment of ROCKi in cultured cells could induce faster senescence after withdrawal of ROCKi treatment. In addition, ROCKi-induced expressions of IL-1β and IL-8 in oral keratinocytes treated with long-term ROCKi were not reversible [52]. These studies suggested that ROCKi treatment is beneficial for effective expansion of keratinocytes, but long-term use of ROCKi in culture may exhaust stem cells, leading to premature senescence and loss of stemness. In our study, a short-term (six-day) ROCKi treatment was used at the beginning of primary keratinocyte culture to achieve increased cell survival and efficiently expand KSCs *in vitro* to avoid the potential senescence. Our results showed that the influence of ROCKi on keratinocytes following a short-term treatment was transient and reversible.

We observed a 15-fold increase in CFE during ROCKi treatment. This was lower than a previously reported 50-fold increase [23], possibly due to differences in tissue origin (the skin from donors with ear reconstruction surgery in this study versus foreskin in Terunuma et al.), culture conditions, and donor-related variability. Despite these differences, we confirmed by related protein markers and single-cell transcriptomic analysis that the accelerated proliferation of primary keratinocytes following six-day ROCKi treatment was due to some holoclone-forming cells transforming to a large number of mero- and paraclone-forming cells. Such an initial boost in keratinocyte growth can save at least 1–2 weeks in the generation of epidermal sheet cultures.

We have noted that ROCKi-treated keratinocytes were migrating non-cohesively with loose cell–cell contacts at the first three days of ROCKi treatment but became tightly packed at day 4. It has been reported that keratinocytes treated with ROCKi had fusiform-like phenotypes (spindle-shaped mesenchymal-like cells) [53] which was similar to cells with depolymerized F-actin [54]. Rho-associated kinase can activate LIMK-kinase 1 (LIMK-1) [55]. As LIMK-1 inactivates the actin depolymerization factor ADF/cofilin binding to F-actin to control actin dynamics, inhibition of Rho-associated kinase by ROCKi can therefore promote activation and binding of ADF/Cofilin to F-actin. This leads to F-actin depolymerization, i.e., loose cell–cell contacts or non-cohesive migration of cells, which we observed in the ROCKi-treated cells. We also detected mesenchymal-like cell morphology at the beginning of ROCKi treatment. Epithelial to mesenchymal transition (EMT) plays an essential role in wound healing. Keratinocytes can undergo EMT during wound healing and stay in this intermediate state (known as mesophase) to meet a high demand of cell proliferation for wound closure [56]. We speculate that isolation and culture of primary keratinocytes *in vitro* creates a situation like wound healing. ROCKi influences the wound healing-like situation by stimulating keratinocytes into EMT through the way of activating F-actin depolymerization and increased adhesion [54,57].

Mitochondrial mass/content increases in stem cells when the stem cells change from resting to activated states [58]. This is due to transition of glycolysis to oxidative phosphorylation (mitochondria-based ATP production), leading to increase in mitochondrial mass/content by the process known as mitochondrial biogenesis [59]. We observed an increased cell population with higher mitochondrial mass/content in ROCKi-treated cells, but we did not detect any changes at the transcriptional level related to mitochondrial biogenesis. Some studies show that, apart from the biogenesis, the inhibition of mitophagy in stem cells could also lead to increased mitochondrial mass [60]. This is because, in somatic cells, mitophagy selectively eliminates damaged mitochondria, whereas in stem cells, mitophagy may serve as a physiologic mechanism for the metabolic rewiring of differentiating stem cells by removing old mitochondria and replacing with new mitochondria [58,59]. We also observed that cells with higher mitochondrial mass and K14 expression were localised in the peripheries of the cell colonies. This indicated that the higher mitochondrial mass cells were at the status of proliferation in which they could relate to the KSCs differentiation to TA cells.

Transcriptomic analysis at single-cell resolution revealed gene expression changes between treated cells and non-treated cells during ROCKi treatment. We detected *VIM* and *F3* differentially expressed in keratinocytes during treatment. Studies by others have also shown that both *VIM* and *F3* are expressed in the skin, in particular during wound healing [47,61]. Although *KRT7, KRT18* and *KRT19* have been reported to be expressed in cultured keratinocytes, their exact biological role *KRT7, KRT18* and *KRT19* in cultured cells is not clear. As they have been considered carcinoma-related markers, the reduction in these genes may at least suggest that ROCKi-induced proliferation is not related to oncogenic transformation [43,44]. We noticed that there was no difference in the protein level of K14 in cells with and without ROCKi treatment. Several potential reasons may be behind this. For example, (i) K14 may not be a specific/unique marker for proliferation cells; (ii) early passaged primary keratinocytes (in our study) may be predominantly comprised of proliferation cells compared to differentiated cells. As abundant K14 were expressed in these cultures, minor change in K14 at protein level could not be detected by immunoblot. We used *ANLN, AURKB*, *CCNA2*, *CKAP2L*, *FOXM1*, *HMGB2* and *LMNB1* as holoclone-forming cell markers, *KRT14, TP63, ITGA6, ITB1, BIRC5* as clonogenic markers, and *SERPINB3, SFN, KRT10, TGM1, IVL, SPINK5* as differentiated cell markers [40] to identify the corresponding populations. We also analysed the rates at which cells differentiate through these cell types. Holoclone-forming cells in ROCKi-treated cell cultures produced mero- and paraclone-forming cells at a faster rate than non-treated cell cultures. Combined with the evidence of decreased holoclone-forming cell proportion and increased transcriptome changes in proliferation-related markers in ROCKi-treated cells, our results suggest that holoclone-forming cells rapidly produced proliferating TA cells (meroclone cells), thereby increasing the cell numbers. This agrees with our observation of increased mitochondrial mass in ROCKi-treated cells. The observed decrease in the proportion of holoclone-forming cells during the ROCKi treatment was due to the increased proportion of proliferating TA cells while the absolute number of holoclone-forming cells were not significantly different between ROCKi-treated cells and control cells. Importantly, transcriptomic analysis further indicated that the proportion of the holoclone-forming cell population in ROCKi-treated cultures was not reduced after the withdrawal of a short-term ROCKi treatment, indicating that the holoclone-forming cells were not exhausted.

We also checked whether cells maintained the ability to terminally differentiate following short-term ROCKi treatment in the 3D culture system. Both non-treated and ROCKi-treated cells were able to develop the epidermal-like structure with multiple layers and expressions of proliferation and differentiation protein markers. Notably, the proliferation marker Δ-P63α was upregulated in the cultures generated using ROCKi-treated cells, but this did not affect the process of terminal differentiation as there were fully developed cornified layers in the 3D culture generated using ROCKi-treated cells. In agreement with others observation, the level of Δ-P63α was similar in the cultures generated using non-treated cells and ROCKi withdrawal cells. This further confirmed the transient effect of short-term ROCKi treatment.

ROCKi plays its biological function via AKT1/2 and ERK1/2 signalling that regulates cell survival, growth, proliferation, and differentiation. The temporal activation of these kinases required to support cellular proliferation and differentiation is cell-type-dependent. Sustained (more than 2 h) ERK activation promoted proliferation in fibroblasts [62], but in neuronal-derived PC12 cells, sustained ERK activation inhibited proliferation, whereas transient activation (15 min) promoted PC12 cell proliferation [62]. Another study showed that the activation of ERK by epidermal growth factor could peak at 10 min but returned to basal level within 30 min [63]. Similar results were observed in HaCat cells where tumour necrosis factor (TNF) induced cell proliferation via ERK activation which peaked at 15 min but returned to basal level within 30 min in the presence of EGF receptor inhibition [64]. We also observed that ROCKi transiently activated both ERK and AKT signalling with the peak activation at 15 min in primary keratinocytes, but returned to basal level within 30 min. It suggests that primary keratinocyte proliferation stimulated by ROCKi is also caused by temporal activation of ERK and AKT, i.e., transient activation (15 min), but not sustained activation of AKT and ERK signalling.

In conclusion, while short-term (6-day) ROCKi treatment improved holoclone-forming cell survival and growth, the influence on keratinocytes was transient, reversible, and did not affect the characteristics of holoclone-forming cells. Our results provide new insights about the effect of ROCKi on human skin epithelial cell proliferation and differentiation. This could open a potential avenue for clinical application by leading to application for regulatory approval and clinical testing which benefits patients with rare genetic skin diseases.

## Figures and Tables

**Figure 1 cells-12-00346-f001:**
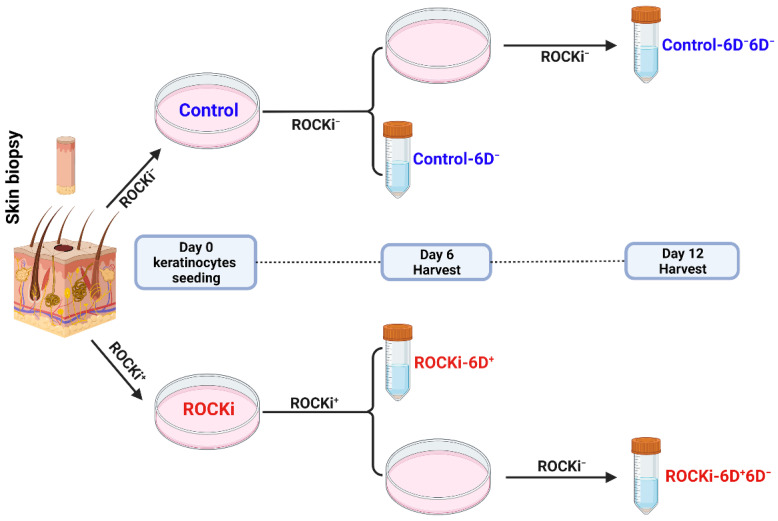
Workflow of experiments and sample collections. Freshly isolated keratinocytes (passage 0) were divided into two populations, either with ROCKi (10 µM Y-27632) or without ROCKi treatment (0.1% DMSO) for six days. Half of the cells were harvested on day 6 and the remaining cells were passaged and cultured without ROCKi for additional six days and then harvested. Image created with BioRender.com.

**Figure 2 cells-12-00346-f002:**
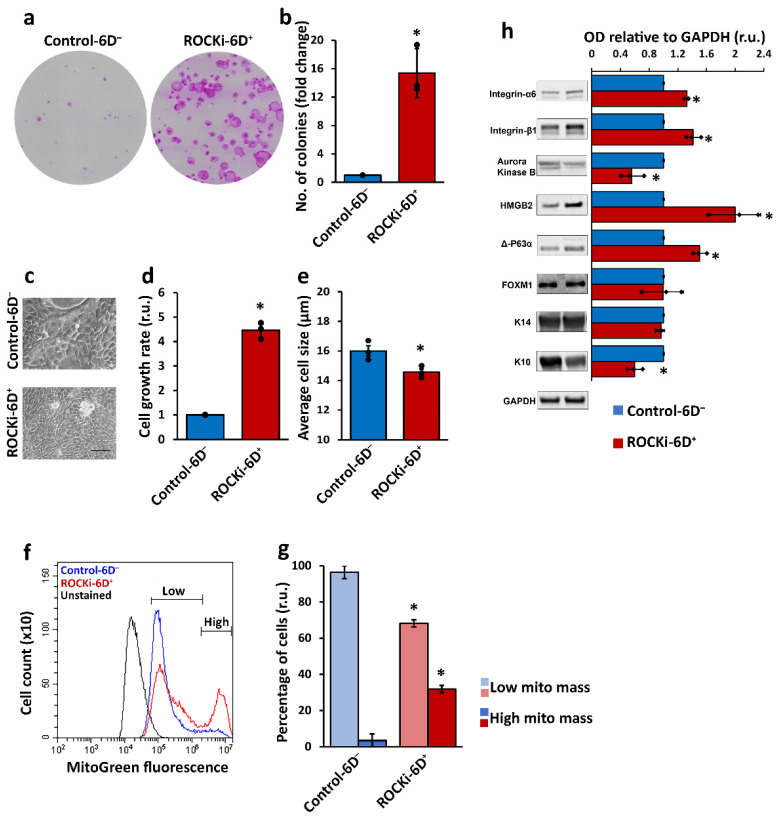
Six-day ROCKi treatment efficiently increased proliferation of primary keratinocytes. Primary keratinocytes at passage 0 were cultured with (ROCKi-6D^+^) and without (Control-6D^−^) ROCKi for six days. Colonies were stained with Rhodanile blue (**a**). A significant increase in colony formation was detected in ROCKi-treated cells (ROCKi-6D^+^) compared to non-treated cells (Control-6D^−^) (**b**). The morphology of cultured keratinocytes at day 6. Scale bar = 50 mm (**c**). Increased cell growth rate in treated cells compared to non-treated cells (**d**). Decreased average cell size in treated cells compared to non-treated cells (**e**). Histogram of mitochondrial mass in treated and non-treated cells stained with 200 nM Mitotracker Green and confirmed by flow cytometry (**f**). Significant increase in cell populations with high mitochondrial mass in treated cells compared to non-treated cells (**g**). Left panel shows the images of representative immunoblots probed with increased proliferation markers and reduced differentiation marker (K10) (**h**). GAPDH was used as a loading control. The right panel shows the densitometry analysis for immunoblots. Data are presented as mean ± S.D., *n* = 3, * = *p* < 0.05.

**Figure 3 cells-12-00346-f003:**
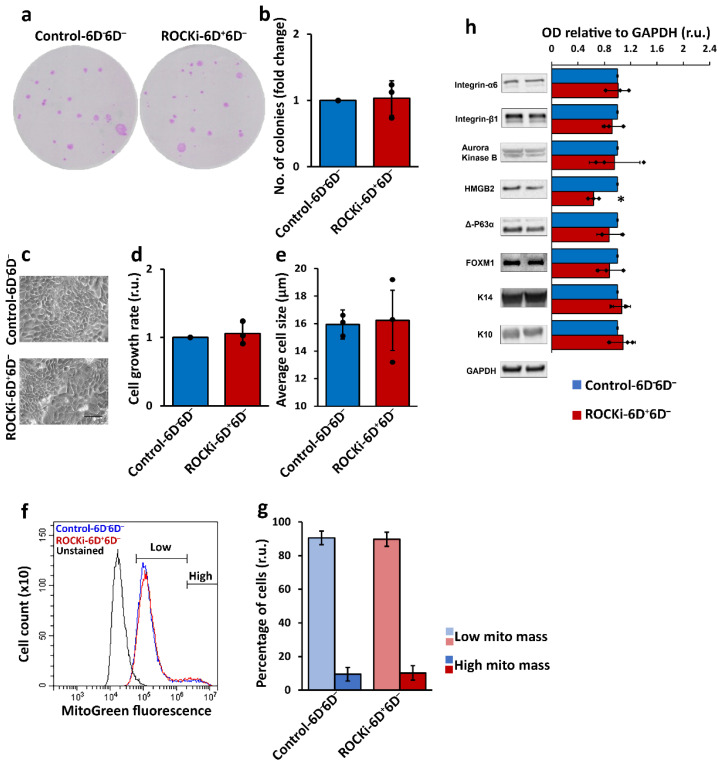
Cells reversed their proliferation status following withdrawal of ROCKi treatment. Control-6D^−^ and ROCKi-6D^+^ cells were passaged once and cultured without ROCKi (Control-6D^−^6D^−^ and ROCKi-6D^+^6D^−^) for additional six days. Colonies were stained with Rhodanile blue (**a**). No significant change in colony formation was detected in treated cells compared to non-treated cells (**b**). The morphology of cultured keratinocytes at day 6. Scale bar = 50mm (**c**). Quantification of the cell growth rate in treated cells compared to non-treated cells (**d**). Quantification of the change in average cell size in treated cells compared to non-treated cells (**e**). Histogram of mitochondrial mass in treated and non-treated cells stained with 200nM Mitotracker Green and confirmed by flow cytometry (**f**). Quantification of the change in cell population with high mitochondrial mass in treated cells compared to non-treated cells (**g**). Left panel shows the images of representative immunoblots with proliferation and differentiation markers with no differences between Control-6D^−^6D^−^ and ROCKi-6D^+^6D^−^ apart from HMGB2 (**h**). GAPDH was used as a loading control. The right panel shows the densitometry analysis for immunoblots. Data are presented as mean ± S.D., *n* = 3, * = *p* < 0.05.

**Figure 4 cells-12-00346-f004:**
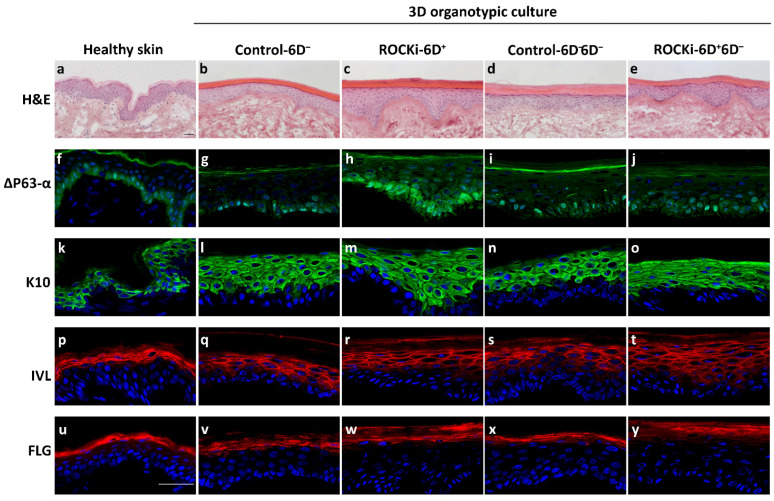
Cells treated with ROCKi for six days had proliferation and differentiation abilities similar to non-treated cells. Three-dimensional organotypic cultures were generated using cells with ROCKi treatment (ROCKi-6D^+^), without ROCKi treatment (Control-6D^−^ and Control-6D^−^6D^−^), and after withdrawing ROCKi treatment (ROCKi-6D^+^6D^−^). The 3D culture morphology examined by Haematoxylin and eosin (H&E) staining (**a**–**e**). Proliferation marker ∆P63-α and differentiation markers K10, involucrin (IVL), and filaggrin (FLG) were examined by immunofluorescence staining (**f**–**y**). Increased ∆P63-α expression was detected in ROCKi-6D^+^compared to other groups of cells, but it was returned to the level to non-treated cells after withdrawal of ROCKi treatment. Nuclei were stained by DAPI (blue). Scale bar = 40 µm. The expression of these proteins in the skin are shown in (**f**,**k**,**p**,**u**).

**Figure 5 cells-12-00346-f005:**
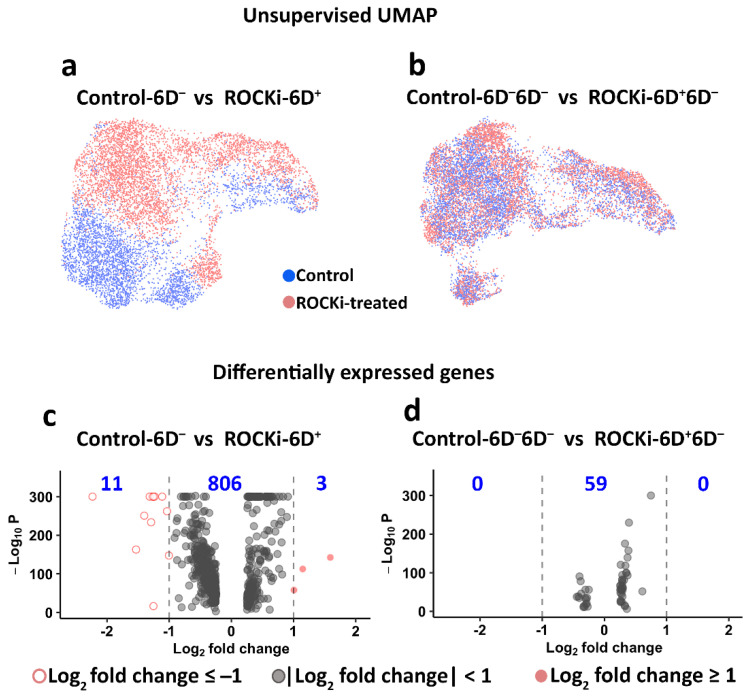
Single-cell transcriptomic analysis. Single-cell transcriptomic analysis was performed using the data generated from scRNAseq. Uniform manifold approximation and projection (UMAP) plots show distinct clusters between non-treated (Control-6D^−^) and ROCKi-treated (ROCKi-6D^+^) cells (**a**), whereas cells from the ROCKi-withdrawn (ROCKi-6D^+^6D^−^) and non-treated (Control-6D^−^6D^−^) groups clustered together (**b**). Differentially expressed genes between ROCKi-treated, ROCKi-treatment withdrawn, and non-treated cells show in the volcano plots (**c**,**d**). The numbers in blue indicate the number of genes.

**Figure 6 cells-12-00346-f006:**
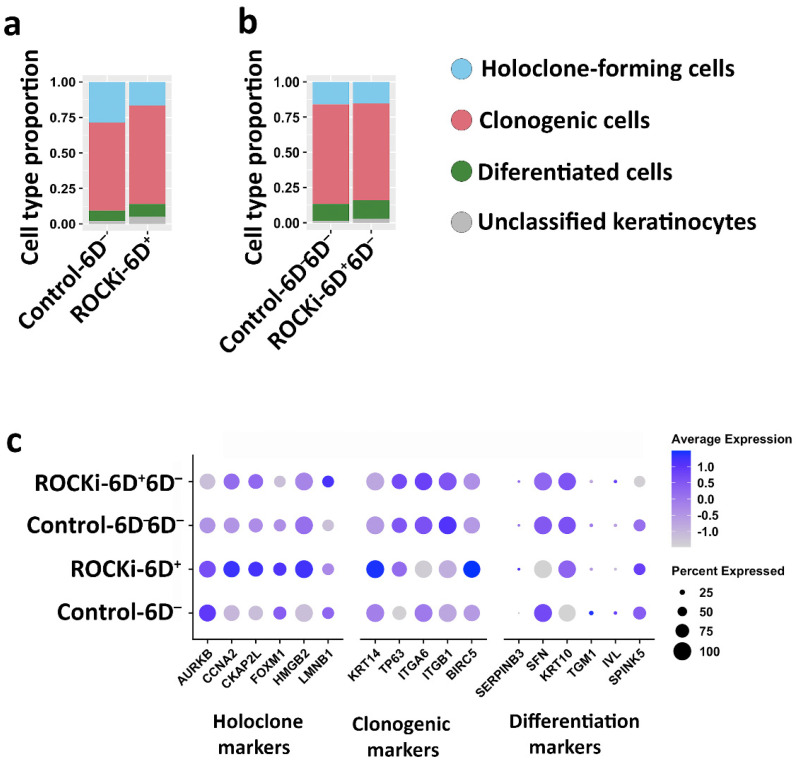
Cell type proportions and gene expressions with and without ROCKi treatment. Cell type populations were analysed using the transcriptomes for holoclone, clonogenic, and differentiated cell markers. Differences in cell type proportions were detected between non-treated cells (Control-6D^−^) and ROCKi-treated cells (ROCKi-6D^+^) (**a**), whereas there was no difference between non-treated cells (Control-6D^−^6D^−^) and ROCKi-withdrawn cells (ROCKi-6D^+^6D^−^) (**b**). Dot plots show the expression of markers for holoclone-forming, clonogenic, and terminally differentiated cells in different cultures (**c**).

**Figure 7 cells-12-00346-f007:**
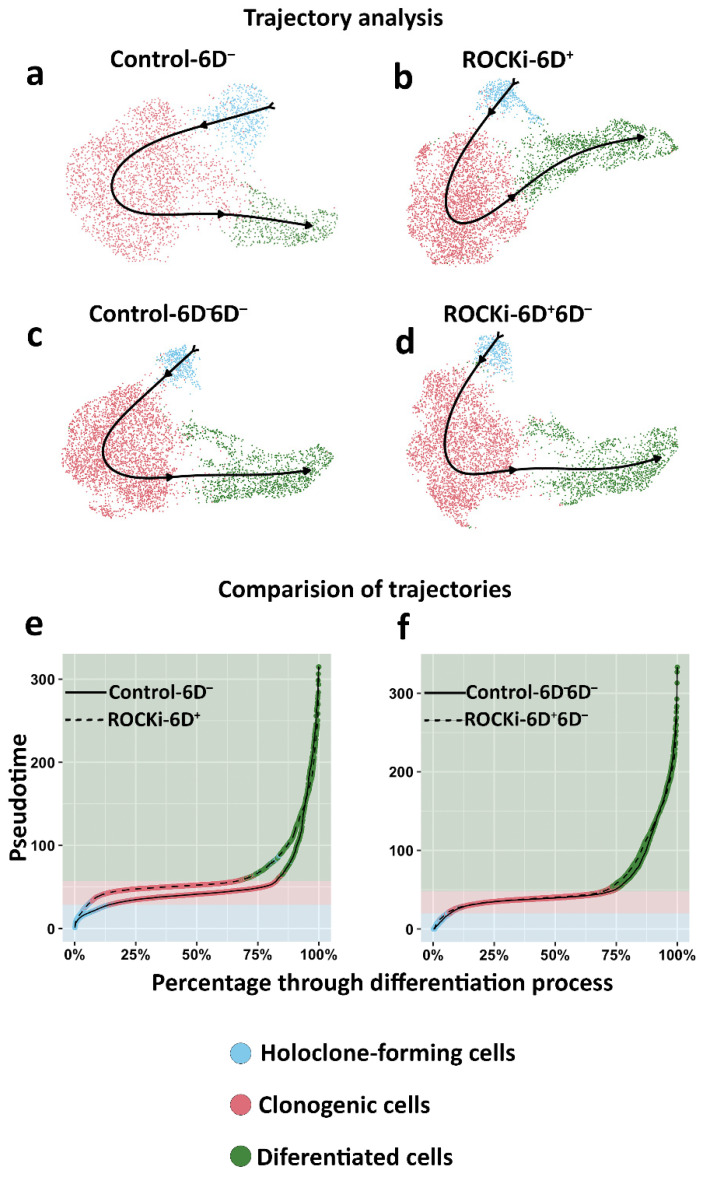
Inferred differentiation trajectories and comparison of pseudotime distributions. Slingshot was used to infer the differentiation trajectories of Control-6D^−^, ROCKi-6D^+^, Control-6D^−^6D^−^, and ROCKi-6D^+^6D^−^ cells, respectively (**a**–**d**). Cell colour corresponds to the three clusters used as input to Slingshot, which correspond to the three cell types. Cumulative distributions of the pseudotimes of Control-6D^−^ and ROCKi-6D^+^ (**e**). Cumulative distributions of the pseudotimes of Control-6D^−^6D^−^ and ROCKi-6D^+^6D^−^ (**f**). The colour surrounding the line in (**e**,**f**) corresponds to the cluster of the cell at that point. The shaded portions in (**e**,**f**) correspond to the dominant cluster at each stage of differentiation.

**Figure 8 cells-12-00346-f008:**
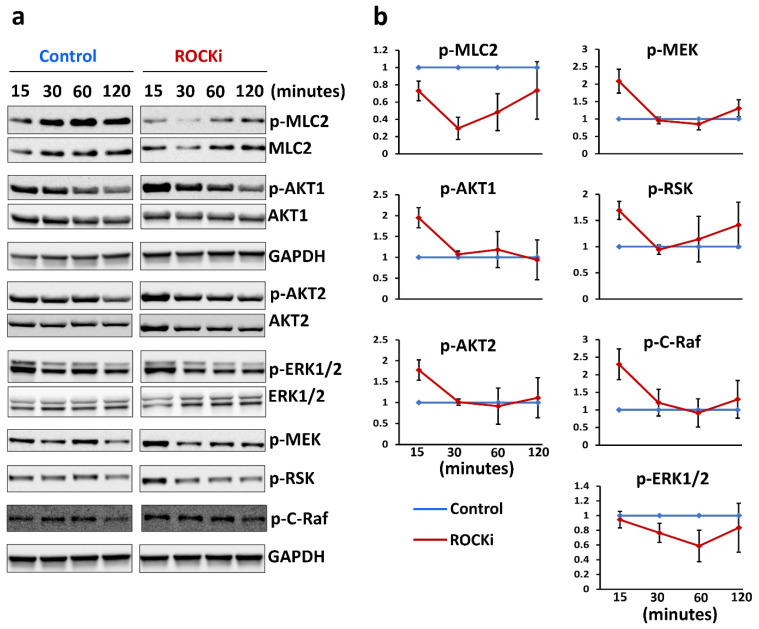
ROCKi-activated AKT and MAPK signalling. Representative immunoblot images show the changes in phosphorylation of kinases involved in AKT and MAPK signalling in cells treated with ROCKi at different time points (**a**). GAPDH was used as a loading control. Densitometry quantification of immunoblot at different time points (**b**). Data were normalized to GAPDH and are presented as mean ± S.D., *n* = 3.

## Data Availability

The R markdown notebook used to generate our single-cell RNAseq results is available at github.com/george-hall-ucl/reversibility_of_rocki_paper_scrnaseq/blob/main/data_analysis_for_paper.md. Single-cell RNAseq data have been deposited in the Gene Expression Omnibus database under accession code GSE207130.

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
