# Peer review of "Short-Term Treatment with Rho-Associated Kinase Inhibitor Preserves Keratinocyte Stem Cell Characteristics In Vitro"

_cells, 2023, doi:10.3390/cells12030346_

Round 1
Reviewer 1 Report
This paper studied the short-term ROCKi treatment of primary keratinocytes to overcome the slow growth of newly isolated keratinocytes that hindered clinical application. And the proposed scheme outperforms the state of the arts and can open a potential avenue for clinical application. There are some problems, which must be solved before it is considered for publication.
1. Relevant research background such as gene therapy and Y-27632 needs to be supplemented in INTRODUCTION, and you should cite all papers you use properly.
2. Discussion needs more in it, as it’s more of afterthought. However, It's more about stating the results than discussing them. For instance, How does ROCKi reduce the loss of stemness of KCSs?
3. The research is innovative, and it would be nice to make the highlights more prominent.
Reviewer 2 Report
Comments on the article cells-21343366 entitled: “Short-term treatment with Rho-associated kinase inhibitor preserves keratinocyte stem cell characteristics in-vitro” By Vignesh Jayarajan
and collaborators.
The Authors tested the hypothesis that Rock inhibitors could improve keratinocyte stem cells maintenance and culture. Towards this end, the authors treated human primary keratinocytes and KSCs isolated from human interfollicular skin with Y-27632. They examined the Y-27632 induced changes on cell growth, protein markers and single-cell transcriptomics. They report that short-term ROCK inhibition induced keratinocytes expansion, and preserved the holoclone-forming cell population (KSCs). The reversal of this treatment makes this study very interesting.
Specific comments
Section Results:
- Figure 2h: In the text line starting: "Most proliferation markers were significantly increased in ROCKi-6D+ cells compared to Control-6D– cells".
However, only 8 markers were tested and only 4 showed an increase.
So, this is not "most markers", hence there is no indication regarding the levels of K14 and K10 in the text.
- Figure 5: the text describing figure 5 does not correspond to the shown figure 5 in this manuscript. The text indicates 11 downregulated genes, but in the figure 5c there are 12 downregulated and 3 upregulated genes. Please clarfy.
- Figure 8: This analysis is not clear. Once can see the changes occurring during the first 15 min but what happens during the next timepoints (30, 60, 120 min) is not explained. Furthermore, the method section corresponding to this figure 8 is missing or unclear where these experiments are described.
Additionally, there are some errors in figure 5. The text describes Figure 5e and 5f, but these panels are missing and should likely be annotated 5c-d.
Method section: Paragraph: Skin samples and primary keratinocyte culture
The composition of the Green’s medium [49] should be added. Note I could not access this article 49.
How was the concentration of ROCKi determined as being used at 10uM?
Did the author perform a cytoxicity assay to determine the concentration of ROCKi? Please clarify.
Overall, this manuscript is interesting and well presented. The majority of the conclusions are supported by the results presented in this manuscript.
Reviewer 3 Report
Jayarajan et al. analyzed the effect of ROCK inhibitor on epidermal culture sheet preparation using Green's method.
The reviewer thinks the content is clear and interesting to readers.
Some minor points are described below.
1. Fig.2h and 3h. The authors show the effect of ROCK inhibitor in immunoblots with protein and comparison to GAPDH.
Since the expression level of GAPDH in keratinocytes is relatively unstable, it is advisable to use several housekeeping genes, if anything. In addition, if possible, better data would be obtained if quantified by real-time PCR.
The reviewer thinks there are seven proliferation markers but not enough analysis of differentiation markers. How about KRT1, FLG, IVL, SPINK5, and LOR?
It is known that expression levels of K14 do not change much before and after the differentiation of keratinocytes. If they are to be used here, I think some references should be added.
2. Fig.6 The result was that adding ROCK inhibitor reduced holoclone.
As the author says, it could work in the right direction in creating epidermal sheets, but the reviewer thinks it depends on the intended use.
For example, if epidermal sheets are to be used for diseases such as epidermolysis bullosa, sheets that retain a large amount of holoclone are needed. The authors should mention this point a little more.
The Holoclone, Mero/paraclone, and Differentiation classifications in Fig. 6c are confusing. Paraclone has a low proliferation potential, so these two (Meroclone and paraclone) should be distinguished.
The first half of this manuscript compares differentiation vs proliferation so that it could be read as (Holo, Mero, para) = differentiation, but the reviewer guesses not.
3. Page 8. The reviewer can't find the figure of Fig 5 e and f. Please correct them to the correct ones.
